# Trade-offs between overall survival and side effects in the treatment of metastatic breast cancer: eliciting preferences of patients with primary and metastatic breast cancer using a discrete choice experiment

Alistair Bullen [1], Mandy Ryan,[2] Holly Ennis,[3] Ewan Gray,[3]
Luis Enrique Loría-Rebolledo [2] Morag McIntyre,[4] Peter Hall[3,4]

¹Edinburgh Clinical Trials Unit, The University of Edinburgh Usher Institute of Population Health Sciences and Informatics, Edinburgh, UK
²Health Economics Research Unit, University of Aberdeen, Aberdeen, UK
³The University of Edinburgh, Edinburgh, UK
⁴Western General Hospital, Edinburgh, UK

**Correspondence to**
Alistair Bullen;
alistair.bullen@gcu.ac.uk

## ABSTRACT

**Objectives** There has been a recent proliferation in treatment options for patients with metastatic breast cancer. Such treatments often involve trade-offs between overall survival and side effects. Our study aims to estimate the trade-offs that could be used to inform decision-making at the individual and policy level.

**Design** We designed a discrete choice experiment (DCE) to look at preferences for avoiding severity levels of side effects when choosing treatment for metastatic breast cancer. Treatment attributes were: fatigue, nausea, diarrhoea, other side effects (peripheral neuropathy, hand–foot syndrome and mucositis) and urgent hospital admission and overall survival. Responses were analysed using an error component logit model. We estimated the relative importance of attributes and minimum acceptable survival for improvements in side effects.

**Setting** The DCE was completed online by UK residents with self-reported diagnoses of breast cancer.

**Participants** 105 respondents participated, of which 72 patients had metastatic breast cancer and 33 patients had primary breast cancer.

**Results** Overall survival had the largest relative importance, followed by other side effects, diarrhoea, nausea and fatigue. The risk of urgent hospital admission was not significant. While overall survival was the most important attribute, respondents were willing to forgo some absolute probability of overall survival for reductions in all Grade 2 side effects (12.02% for hand–foot syndrome, 11.01% for mucositis, 10.42% for peripheral neuropathy, 6.33% for diarrhoea and 3.62% for nausea). Grade 1 side effects were not significant, suggesting respondents have a general tolerance for them.

**Conclusions** Patients are willing to forgo overall survival to avoid particular severity levels of side effects. Our results have implications for data collected in research studies and can help inform person-centred care and shared decision-making.

## STRENGTHS AND LIMITATIONS OF THIS STUDY

⇒ Our study employs a discrete choice experiment methodology which is capable of estimating trade-offs for metastatic breast cancer treatment in accordance with economic utility theory.
⇒ The selection of attributes was informed by a broad selection of work packages employing qualitative methods and reviewing a variety of literature.
⇒ We estimated the trade-offs between overall survival and symptoms and side effects of fatigue, nausea, diarrhoea, peripheral neuropathy, hand–foot syndrome and mucositis.
⇒ We cannot include all attributes that determine the choice of treatment.
⇒ Due to recruitment difficulties we include patients with both primary and metastatic breast cancer; these patients may have different preferences.

## INTRODUCTION

There are 35 000 people in the UK living with metastatic breast cancer (mBC).[1] mBC occurs if the cancer spreads to another part of the body at which point the cancer is usually considered incurable. The focus of treatment then shifts from curing the disease to managing it, slowing further progression and palliating symptoms. There is a dichotomy at the core of discussions surrounding treatment in this context, namely the trade-off between overall survival (OS) and the side effects patients must tolerate.[2] Different treatments offer variable prospects for survival versus side effects. Treatment decisions are made more complex by the proliferation of new medicines for the treatment of mBC, ranging from cytotoxic chemotherapy to hormone therapies. Recent new

additional options include immunotherapy and targeted small molecules.[3]

Such developments mean that patients with breast cancer must navigate difficult decisions between complex and unfamiliar treatments.[4] Greater patient involvement in decision-making is needed to allocate the treatment that best addresses their needs. Recent guidelines have emphasised the requirement for shared decision-making across the National Health Service.[5 6] Although shared decision-making is widely practised its implementation needs improvement, specifically regarding doctor–patient communication.[7] Evidence from patient preference studies reveals trends to be considered by healthcare providers during consultations. Patient preferences are also important for the authors of healthcare guidelines that inform policy around which drugs should be provided. As a final example, they are important for developers of new cancer drugs when they provide guidance on what patients will tolerate concerning side effects for improvements in survival.

Discrete choice experiments (DCEs), sometimes referred to as conjoint analysis, are increasingly used to estimate patient preferences, looking at the relative importance of attributes as well as the trade-offs individuals are willing to make.[8] A recent systematic review of the application of DCEs to oncology treatment identified 79 studies, with patient preferences for breast cancer (n=10, 13%) as the most common area of application.[9] The review found the most common outputs were relative importance of attributes and marginal rates of substitution (MRS, trade-offs) in terms of (in order of frequency): willingness to pay (WTP), minimum acceptable benefit, minimum acceptable risk and willingness to accept non-risk for benefit and willingness to travel. While clinical efficacy attributes were commonly ranked as most important, with OS and progression-free survival (PFS) ranked most important by 90% and 30%, respectively, by patient samples across all cancer types, respondents were often willing to trade clinical efficacy for improvements in side effects. A similar result was found in a systematic review of patient preference studies relating to breast cancer treatment.[10] These two systematic reviews identified six DCEs that assessed preferences for mBC drug treatments.[11–16] These studies also show that while treatment efficacy (OS or PFS) is important, and often the most important factor, patients also value avoiding the side effects of different treatments.[11 14–16] Two of these mBC studies estimated the value of avoiding side effects in monetary terms (WTP, a monetary measure of benefit).[13 14] We use the DCE methodology to investigate how much absolute probability of OS people are willing to give up to avoid a particular severity level of side effects in the treatment of mBC. We refer to this as minimum acceptable survival (MAS). We also focus on the severity of side effects, whereas the existing DCEs have focused mainly on the risk of side effects, and the preferences for long-term survival. Our study is also the first to elicit preferences for the treatment of mBC in the UK; preferences across countries may differ due to cultural factors and different healthcare systems. For example, Southeast Asian attitudes to cancer management and death are known to be different from Western ones.[17]

## METHODS

The DCE is a choice-based survey that quantifies preferences for alternatives (eg, treatment options for mBC) where alternatives are described by their attributes and associated levels.[18] In our DCE alternatives are treatments, attributes are treatment characteristics (eg, survival and side effects), and levels are values associated with treatment characteristics (eg, % chance of survival, possible levels of severity for nausea).

### Defining attributes and levels

Four work packages (WPs) informed the attributes and levels: (1) a targeted literature review of qualitative literature concerning the patient experience of metastatic cancer, (2) a targeted literature review of DCEs centred on treatments for metastatic cancer, (3) a thematic analysis[19] of Scottish Medicine's Consortium Patient and Clinical Engagement statements for mBC treatments and (4) face-to-face interviews with patients with mBC. All work involving face-to-face patient contact was completed by a research nurse and research assistant both of whom had been trained in qualitative methods. For more information on all WPs see online supplemental file 1. The research group, consisting of breast cancer and DCE experts, considered these attributes, reducing them to a manageable number for use in the DCE framework. Attribute selection and layperson definitions were developed using think-aloud interviews with patients.[20]

The final attributes and levels are shown in table 1, with patient definitions of attributes defined in table A1 in online supplemental file 2. Levels are intended to represent possibilities for first-line treatment following a diagnosis at Stage IV (mBC). Side effects were: fatigue, nausea, diarrhoea and additional side effects (peripheral neuropathy, hand–foot syndrome and mucositis as mutually exclusive levels). Levels of side effects attributes were described using plain-language translations of the Common Terminology Criteria for Adverse Events (CTCAE)[21] criteria (online supplemental table A1). These were developed with health professionals and tested in the developmental piloting work. Following piloting with patients, and to ease understanding, fatigue was referred to as tiredness. The nausea attribute combined the corresponding CTCAE grades nausea and vomiting (since they tend to accompany one another). Attribute levels ranged from a zero level of toxicity up to Grade 2. Choice options were discussed with health professionals to ensure plausibility. During these discussions it was suggested that some background fatigue is expected for most patients; therefore Grade 1 fatigue was the minimum level of the attribute. It was also advised that in the presence of Grade 3 adverse events, treatment would be discontinued; thus,

**Table 1** Attributes and levels for the discrete choice experiment

| Attributes | Levels | Definition | Regression equation label | Regression equation preference parameter |
|---|---|---|---|---|
| Fatigue* | Grade 1 Fatigue (reference level)<br>Grade 2 Fatigue | Tiredness – In this scenario your cancer will always make you more tired than you once were. But treatments can make this worse | G2_FAT | $\beta_1$ |
| Nausea | No nausea (reference level)<br>Grade 1 Nausea<br>Grade 2 Nausea | Treatments may cause nausea and nausea may cause you to vomit. | G1_NAU<br>G2_NAU | $\beta_2$<br>$\beta_3$ |
| Diarrhoea | No diarrhoea (reference level)<br>Grade 1 Diarrhoea<br>Grade 2 Diarrhoea | Treatments may cause diarrhoea. | G1_DIA<br>G2_DIA | $\beta_4$<br>$\beta_5$ |
| Additional side effects | No other side effects (reference level)<br>Grade 2 Peripheral Neuropathy<br>Grade 2 hand–foot syndrome<br>Grade 2 Mucositis | A treatment may be associated with an additional side effect. These side effects include peripheral neuropathy (nerve damage), hand–foot syndrome (severe skin problems), and mucositis (mouth ulcers). You can experience a maximum of one of these side effects on a given treatment. | G2_NEU<br>G2_HAN<br>G2_MUC | $\beta_6$<br>$\beta_7$<br>$\beta_8$ |
| Overall survival | 60 alive at 1 year, 8 alive at 5 years<br>65 alive at 1 year, 12 alive at 5 years<br>75 alive at 1 year, 24 alive at 5 years | How long someone lives is always uncertain but in this scenario the care team is able to tell you how many patients are expected to be alive after 1 and 5 years. They are also able to tell you how many of those who survived | OS | $\beta_9$ |
| Risk of urgent hospital admission | 1/100 people<br>10/100 people<br>30/100 people | the first year also experienced an urgent hospital admission. A patient may, for example, have an urgent hospital admission because of a severe infection (sepsis) or because of extreme symptoms. Hospital admission and survival statistics will both be presented in a single graphic. Please imagine that the figure for urgent hospital admissions includes hospital stays which range from days to weeks. | UHA | $\beta_{10}$ |

*Following piloting, and to ease understanding, fatigue was referred to as tiredness.

the maximum level for all adverse event attributes was Grade 2. The additional side effects attribute was included to capture a broader range of side effects while limiting the number of attributes and therefore the cognitive burden of completing the choice tasks.[22] It differed from competing attributes due to each level corresponding to a unique side effect, Grade 2 descriptions were used so that we could compare preferences for the equivalent highest level of the diarrhoea and nausea attributes.

### Patient and public involvement

Patients with mBC were invited to, and participated in, interviews and in-person questionnaire piloting sessions, both of which informed the final design of the survey.

A risk of urgent hospital admission (UHA) was included, defined as the number of people from 100 treated who would be admitted to the hospital for a UHA. The decision to make UHA a probabilistic attribute was motivated by discussions with health professionals. It was suggested that, unlike Grade 1 and Grade 2 toxicities, a treatment that guaranteed a UHA would not be offered to patients. OS was defined as the annual probability of survival, which was time constant and represented the probability of surviving in the present and future years. To account for short-term and long-term preferences[23] annual probability of survival was presented as frequencies at 1 and 5 years, for example, 65% translated to 65 people alive at 1 year and 12 alive at 5 years (the rounded result of $100 \times 0.65^5$). Risk is generally not well understood by the general public,[24] therefore 1-year and 5-year survival were presented alongside one another to illustrate the effects of cumulative probability on respondents. The average 1-year survival rate after diagnosis for a patient with mBC is approximately 65%[25]; we chose this as our central value for our annual survival rate. We used an exponential calculation for 5-year survival, rather than real-world data, to simplify the choice task to include only one risk attribute. The levels for UHA were defined following discussions with health professionals.

It was observed during piloting that some of the expected negative preferences for UHA would occur due to a risk of death. Respondents often struggled to disentangle and interpret the related attributes. To isolate the effect independently from the risk of death a graphic was devised, which showed levels of both attributes. The combination of frequencies and tree diagrams has been shown to improve understanding of risks.[26 27] The first row reports the number of patients admitted to the hospital for a UHA, and the second and third show 1-year and 5-year survival, respectively. Frequencies for positive outcomes (no hospitalisation and survival) and negative outcomes (hospitalisation and death) were both communicated in an attempt to address framing bias.[28]

### Choices presented to individuals

Ngene (Choice Metrics) was used to create a set of choices from which preferences could be estimated for all possible scenarios; the design was D-efficient, which minimised the variance-covariance of the measures of average preference.[29] This resulted in a set of 12 choice tasks. All choices included a no-treatment option, with side effects defined as the least severe level and risk of UHA 0%. To define the opt-out level of survival respondents were asked what they perceived their chances of survival at 1 and 5 years, resulting in a 45% average level. This was consistently lower than all levels of OS with treatment and judged reasonable given survival at 1 year among patients with stage 4 breast cancer diagnosed in England in 2013 was between 16–43% depending on age, with a mode of 43%.[30] The choice context was described in terms of the scenario, the treatments and side effects as follows:

- ► *The scenario: You are being asked to consider the decision you would make if presented with different metastatic breast cancermBC treatments. For each question there are only 2 treatment options. If you choose a treatment, the other treatment will not be an option tofor you in the future. We ask you to imagine that no other treatment options will become available to you in the future. You also have the option to choose to have no treatment. With no treatment you would experience the symptoms of your cancer; your cancer will be left to progress and you will have shorter life expectancy as a result.*
- ► *The treatments: Both treatments are in the form of daily pills. Both treatments can treat you for the rest of your life. You would be allowed to stop treatment whenever you wished. Both treatments have different benefits and side effects.*
- ► *Side effects: Side effects are guaranteed. Side effects are already being managed with the best available medicines and care. You will still experience a side effect for weeks at a time.*

Following developmental work, the 12 choices were divided into two blocks of six choice tasks to mitigate mental fatigue effects.[31] Respondents were randomly allocated to one of the design blocks and choice tasks were presented in a randomised order. Respondents were given a warm-up choice task (figure 1) to complete.

### Data analysis

The following utility/benefit function was estimated using Error Component Mixed Logit regression:

$$U_{in} = \beta_0 Treat_i + \beta_1 G2\_FAT_{i1} + \beta_2 G1\_NAU_{i2} + \beta_3 G2\_NAU_{i3}$$
$$+ \beta_4 G1\_DIA_{i4} + \beta_5 G2\_DIA_{i5} + \beta_6 G2\_NEU_{i6} + \beta_7 G2\_HAN_{i7}$$
$$+ \beta_8 G2\_MUC_{i8} + \beta_9 OS_{i9} + \beta_{10} UHA_{i10} + \varepsilon_{in}$$

$U_{in}$ represents the utility for individual n for alternative i. The attribute variables are defined in table 1. $\beta_1$ to $\beta_8$ are modelled as dummy variables, showing the value of that attribute level relative to the reference (best) level. $\beta_9$ and $\beta_{10}$ are modelled as continuous variables, showing the value of a % change in OS and UHA. The signs of the $\beta$ parameters indicate whether the effect of the attribute level on preference is positive or negative. All side effects preference parameters are expected to have a negative sign relative to the reference level. Respondents are expected to prefer higher OS, resulting in a positive $\beta_9$. The preference for a chance of UHA, $\beta_{10}$, is expected to have a negative sign, with lower values preferred. $\varepsilon_{in}$

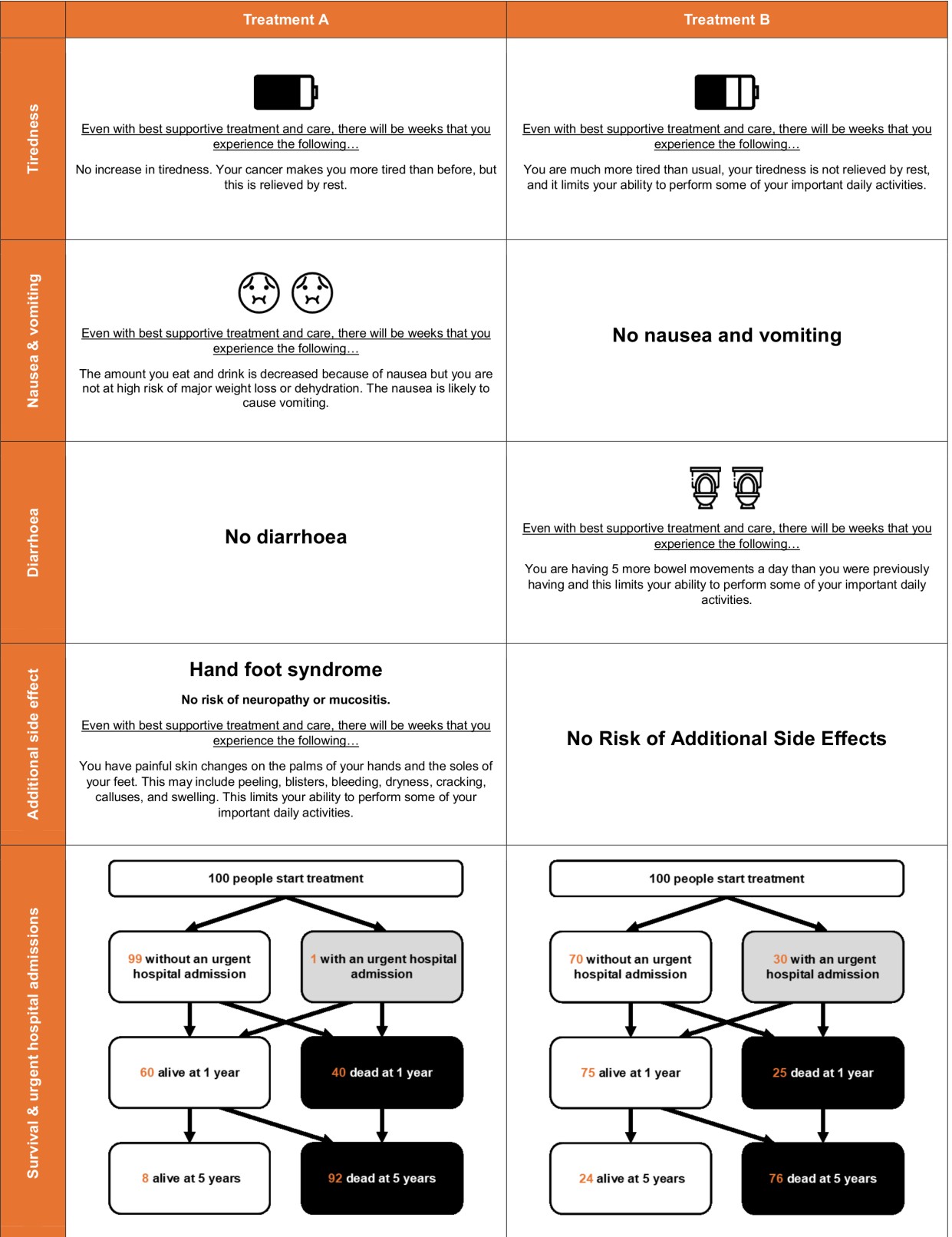

**Responses for Patients:**

Which option would you choose?

○ I would choose to take treatment A
○ I would choose to take treatment B
○ I would choose neither. I understand that I would have worse expected survival as a result.

**Figure 1** Example of discrete choice experiment choice task (warm-up task).

**Table 2** Error component logit

| | | Estimate | P value | 95% CI lower bound | 95% CI upper bound | Relative attribute importance | Minimum acceptable survival |
|---|---|---|---|---|---|---|---|
| Alternative specific constant | Treatment | 5.1339 | 0.0002 | 2.4566 | 7.8112 | – | – |
| | SD of treatment | 4.0982 | 0.0001 | 2.0411 | 6.1553 | | |
| Fatigue | Grade 2 fatigue | −0.2948 | 0.0101 | −0.5194 | −0.0702 | 0.0590 | 2.5419 |
| Nausea | Grade 1 nausea | −0.4196 | 0.0519 | −0.8426 | 0.0034 | 0.1091 | 3.6178 |
| | Grade 2 nausea | −0.5446 | 0.0093 | −0.9550 | −0.1342 | | 4.6960 |
| Diarrhoea | Grade 1 diarrhoea | 0.0241 | 0.8806 | −0.2898 | 0.3379 | 0.1519 | −0.2074 N.S. |
| | Grade 2 diarrhoea | −0.7343 | 0.0004 | −1.1384 | −0.3302 | | 6.3314 |
| Additional side effects | Grade 2 peripheral neuropathy | −1.2087 | 0.0000 | −1.6458 | −0.7715 | 0.2793 | 10.4211 |
| | Grade 2 hand–foot syndrome | −1.3946 | 0.0000 | −1.8404 | −0.9489 | | 12.0247 |
| | Grade 2 mucositis | −1.2764 | 0.0000 | −1.6668 | −0.8861 | | 11.0055 |
| Overall survival | Annual probability of survival | 0.1160 | 0.0000 | 0.0847 | 0.1473 | 0.3485 | – |
| Urgent hospital admission | Probability of urgent hospital admission in the first year of treatment | 0.0090 | 0.1068 | −0.0019 | 0.0199 | 0.0522 N.S. | −2.3236 N.S. (for 30% level) |
| **Model statistics** | | | | | | | |
| Number of individuals | 105 | | | | | | |
| Observations | 601 | | | | | | |
| Log likelihood | −379.9434 | | | | | | |
| Bayesian info criterion | 836.6699 | | | | | | |

N.S, not significant.

represents the unobserved error component. $\beta_0$ shows the general preference for treatment over no treatment (everything else equal) with a positive sign indicating a general preference to receive treatment (everything else equal). An error component is assumed by specifying $\beta_0$ as random normally distributed, thus allowing for flexible substitution between alternatives and dropping the irrelevant alternatives assumption,[32] we run 100 draws using the Halton sequence.

We used the parameter values to estimate the relative importance of attributes[33]; this is calculated as the difference in the range of attribute's variable values. We calculate percentages from these relative ranges, obtaining a set of attribute importance values that add to 100%. We also estimate MRS in the form of MAS for improvements in side effects using the rate for 1-year OS in the calculation, estimated as $\frac{\beta_x}{-\beta_9}$. For example, $\frac{\beta_1}{-\beta_9}$ shows MAS for a reduction in side effects from Grade 2 fatigue to Grade 1 fatigue and $\frac{\beta_4}{-\beta_9}$ shows MAS for a reduction in side effects from Grade 1 diarrhoea to no diarrhoea.

## Sample and recruitment

Calculating an optimal sample size for newly designed DCEs is problematic as it depends on the true values of

the unknown parameters for which the analysis intends to estimate.[34] Previous DCEs in the area of metastatic cancer of a similar design have demonstrated that reliable analysis can be performed with samples of 100 or fewer participants.[35–37] We therefore aimed to recruit 100 patients as a minimum threshold.

We planned to recruit a sufficient number of people with experience of mBC to exceed the minimum threshold. Given the anticipated challenges of recruiting a sufficient number of people who had an mBC diagnosis, the original protocol also included the collection of responses from people who had experienced primary breast cancer. Respondents who responded that they had only a primary breast cancer were asked to imagine that they had received a secondary breast cancer diagnosis in the introductory text. The preferences of patients with mBC were compared with patients with primary breast cancer.

The DCE was administered using an online link between January and March 2020. Recruitment methods included: (1) distribution of leaflets at cancer centres and conferences, (2) an online panel provided by Dynata, (3) social media engagement with help from breast cancer charities and (4) a research nurse approaching patients

directly during clinic visits and inviting them to complete the survey on a tablet device. Interviewed respondents provided informed written consent before interviews proceeded. Access to the survey was unrestricted for people who had acquired the link. Patients self-identified as having had a primary or mBC diagnosis at some point, being a UK resident and 18+ years of age. Inclusion in the sample was not restricted by gender.

## RESULTS

The sample size was 105 (table A2 in online supplemental file 2). All identified as women. 72 respondents were patients with mBC and 33 were patients with primary breast cancer.

10 respondents did not complete all 6 choice tasks, resulting in 29 missing choice tasks. Completed choice tasks were included in the analysis. Of 601 responses to choice tasks across all participants, 38 (6.32%) were for no treatment. These were selected by 16 women, with 3 women always choosing the opt-out option. 32.38% (N=34) of respondents always chose the option with the highest OS. Some of these respondents may have been using a simplifying heuristic, nonetheless, we focus our analysis on the complete sample as it is not possible to distinguish respondents who are demonstrating

a genuine preference and those using a simplifying heuristic. (figures A1 and A2 in online supplemental file 3 compare analyses when excluding the 34 potential non-traders; as expected the relative importance of OS is lower and participants have a higher MAS. However, samples are too small to demonstrate statistically significant differences.)

Table 2 shows the error-component logit regression results for all respondents (table A3 in online supplemental file 2 shows the results of the equivalent multinomial logit) and figure 2 shows the relative importance of attributes. We also ran an alternative specification as multinomial logit where the OS attribute was dummy coded and it demonstrated a near linear relationship between effect and survival gain between the 60 and 75 levels which suggests the specification of OS as a constant variable is appropriate (table A4 in online supplemental file 3).

MAS estimates (table 2, column 8 and figure 3) show respondents' willingness to forgo OS to avoid all Grade 2 toxicities.

Results comparing patients with mBC and patients with primary breast cancer are shown in figures A3 and A4 in online supplemental file 3. The most notable difference is the estimated importance of the nausea attribute,

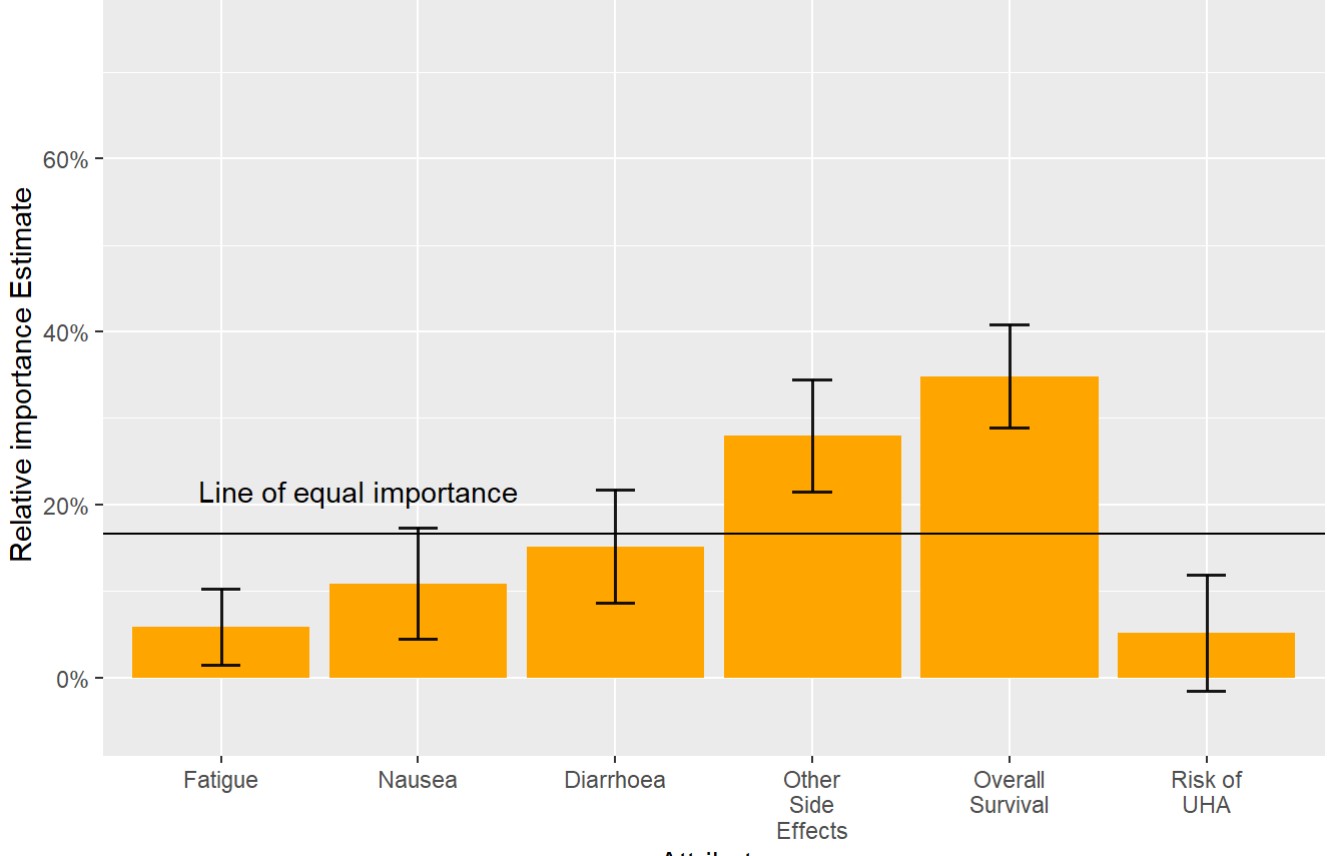

**Figure 2** Relative importance of attributes. Error bars show 95% CI using delta method SEs. UHA, urgent hospital admission.

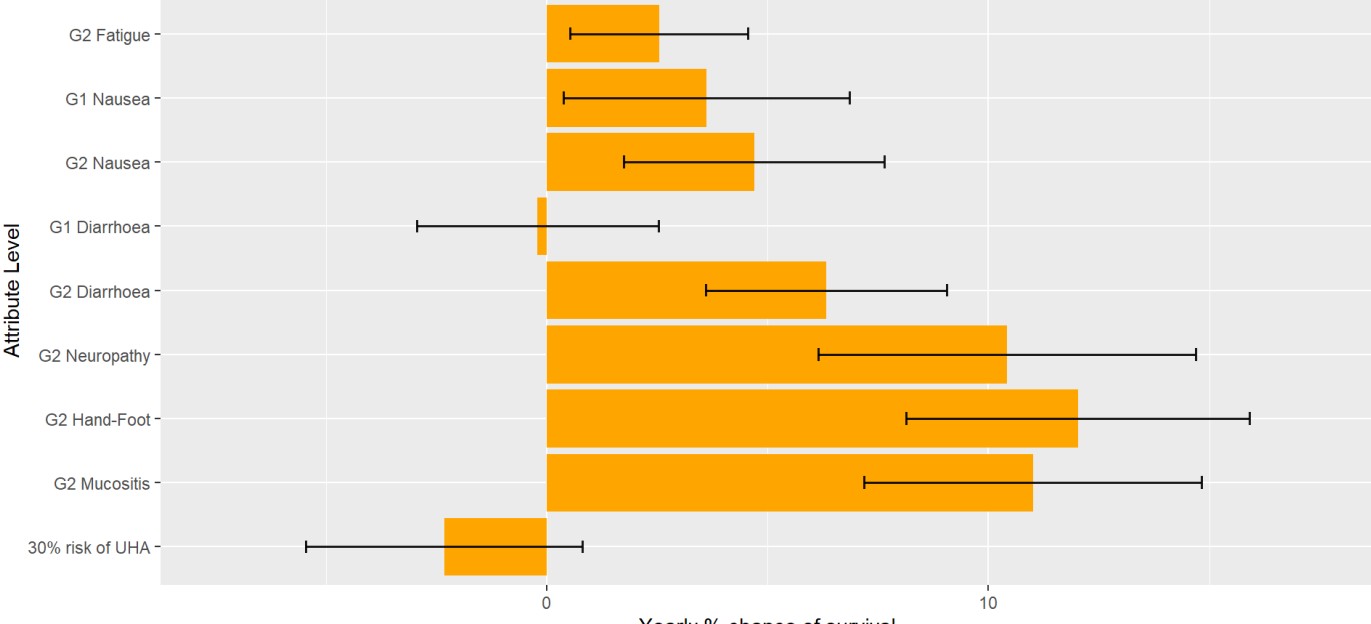

**Figure 3** Minimum acceptable survival to avoid side effects. Error bars show 95% CI using delta method SEs.UHA, urgent hospital admission.

nonetheless, there are no statistically significant differences between any of the estimates.

## DISCUSSION

We provide new evidence on UK women's preferences for the treatment of mBC. Respondents had a general preference for treatment, indicated by the low opt-out rates which result in a positive constant term (Treat). As expected, they preferred treatments with higher OS, in fact almost one-third of the sample (32.38%) always chose the treatment option with a higher OS. All Grade 2 toxicities were significant and negative, suggesting negative preferences for these attribute levels. However, Grade 1 nausea and diarrhoea were not significant, suggesting patients are indifferent when compared with having none of these side effects. There was no significant effect of UHA on respondents' choices.

The relative importance of OS exceeded all other attributes, with an overall importance score of 34.85%. The remaining relative importance was distributed accordingly: additional side effects (27.93%), diarrhoea (15.19%), nausea (10.90%), fatigue (5.90%) and risk of urgent hospital admission (5.22%). Respondents would accept a reduction in the probability of survival of 2.54% to avoid Grade 2 fatigue (and have Grade 1 fatigue). The MAS associated with levels of the additional side effects were particularly high: respondents were willing to give up 10.42%, 12.02% and 11.01% chance of OS for total avoidance of Grade 2 peripheral neuropathy, Grade 2 hand–foot syndrome and Grade 2 mucositis, respectively. Notably, Grade 1 nausea and diarrhoea were acceptable to patients and did not significantly impact patients'

choices. Thus, they were not willing to give up survival for improvements in such Grade 1 side effects. However, Grade 2 side effects were disliked and respondents were willing to forgo up to 12.02% OS to avoid such severe side effects.

Our results add to a growing literature showing that patients with breast cancer value avoiding the side effects of treatments, and are willing to forgo some level of treatment efficacy to achieve this.[9 10] Directly comparing preference estimates between studies is often inappropriate as estimates only apply to the attributes and levels within the choice framework of DCE from which they are derived. Nonetheless, it is important to highlight the findings of other studies and draw comparisons where appropriate. Our results appear to align somewhat with DiBonaventura et al's.[11] exploration of the preferences of women with mBC in the USA who also found that OS was the most important attribute. Additionally, side effects (alopecia, fatigue, neutropenia, motor neuropathy and nausea/ vomiting) and dosing regimen were also important. The remaining studies did not include attributes for OS but did identify statistically significant preferences for side-effect avoidance. For example, Omori et al[15] explored the preferences of Japanese postmenopausal patients with hormone receptor-positive breast cancer for the treatment of mBC. They conclude that women preferred treatments that extend PFS despite potential Grade 2 diarrhoea. However, when diarrhoea severity increased to Grade 3, patients were more willing to sacrifice PFS to avoid more frequent diarrhoea. In contrast, exploring preferences of women diagnosed with mBC in Germany, Spaich et al[16] concluded that severe neutropenia was the

most important attribute, followed by alopecia, neuropathy and PFS. Two studies have explored the preferences of women diagnosed with mBC in the USA, estimating value in monetary terms. Lalla et al[12] found that women were willing to pay the most to avoid severe diarrhoea (US$3894 a year), followed by avoidance of hospitalisation due to infection (US$3279), severe nausea (US$3211) and severe peripheral neuropathy (US$2764). MacEwan et al[13] found that women were willing to pay US$1930 per month for treatment, with US$63 per month for each 1% reduction in the risk of moderate-to-severe side effects. In a similar study in Thailand, Ngorsuraches and Thongkeaw[14] found respondents were willing to pay US$151.6 per month for every 1 month increase in PFS compared with US$69.8 and US$278.3 per month for every 1% decreased risk of anaemia and pneumonitis, respectively.

Our results imply that treatment efficacy and OS are not the only endpoints of value to women with mBC (and indeed oncology more broadly). Furthermore, there is evidence that the CTCAE grading criteria do not scale in parallel with patients' preferences; for example, Grade 2 nausea is preferred to Grade 2 hand–foot syndrome (indicated by a lower negative preference parameter). Grade 1 toxicities were not significant, suggesting they are relatively tolerable to patients (compared with having no side effects). These findings suggest that clinician-reported and objectively graded toxicities may not correspond to patients' values and support the further incorporation of patient-reported outcomes (PROs) and preference studies in the study of new medicines for mBC. PROs are increasingly accepted by the US Food and Drug Administration and European Medicines Agency[38] and the National Institute for Health and Care Excellence has begun to accept patient preference studies alongside traditional evidence such as cost per quality-adjusted life year.[39]

Our study has focused on the preferences of patients. Given that health professionals often make treatment decisions/recommendations for patients, a fruitful area for future research is to compare the preferences of patients and doctors. Current research suggests that it is common for there to be a mismatch in the preferences of patients and healthcare providers.[40] Given health professionals possess greater information on treatments and patients possess private information on their values and priorities, decision aid tools (DATs) can help understand and bridge this mismatch as part of shared decision-making. The focus of such DATs within breast cancer has been on the detection and prevention of early breast cancer.[41] The work presented in this paper contributes to the groundwork for the use of a DCE as a DAT to promote shared decision-making and person-centred care. A limited number of studies have adapted DCEs into DATs: Dowsey et al[42] used a DCE as part of a decision aid for patients undergoing total knee arthroplasty; Hazlewood et al[43] evaluated a proof-of-concept DAT for patients with early rheumatoid arthritis, which included

a DCE to assist respondents in choosing initial treatment and Loría-Rebolledo et al[44] are exploring the use of DCEs to estimate preferences at the individual level for use in a shared decision-making setting.

There are limitations to this study. First, the sample size was small, and we were required to supplement the patient with mBC sample with primary breast cancer who were asked to imagine a secondary diagnosis. Although the analysis did not present large enough differences in preferences to suggest this meaningfully affected results, a larger sample would allow the possibility of preference heterogeneity to be extensively explored. Preferences, trade-offs and willingness to avoid particular side effects may be influenced by many factors. One potential area for future research is understanding the dynamics of treatment preferences and response shift. This may be particularly important for end-of-life care, which patients with mBC may face.[45] Other factors that may influence preferences include specific cancer diagnosis, location of metastases, multiple diagnoses and treatment experience. Future research should collect data on the characteristics of respondents which could be used to explore preference heterogeneity. Second, national data indicates that the highest incidence of new breast cancer cases (any stage) for women between 2015 and 2017 was aged 60–69,[46] suggesting our sample is younger with the largest group aged 50–59. A 2008 survey in the USA found a stronger preference for quality of life than quantity of life among patients with cancer,[47] if this effect exists in our population the preference weights may be positively skewed. Third, the argument could be made that the description of how side effects are experienced in the choice scenario may be difficult for patients to understand. The decision to focus on symptom severity and to avoid clear definitions of symptom frequency relating to side effects was made to alleviate the cognitive burden of the task by simplifying the information presented. We opted to represent uncertainty by suggesting that treatments were indefinite and side effects would therefore be indefinitely experienced 'for weeks at a time'. Some would argue that in doing so we forgo a degree of clarity of interpretation for respondents and consequently the results of the study. Fourth, we simplified the choice task to include only one risk attribute, we used an exponential function to estimate the 5-year survival rate. Future research could include two attributes, 1-year and 5-year survival, with the latter based on real data. Preferences for short-term and long-term survival could then be estimated. Fifth, in defining the no treatment option, the level for OS was defined as the mean value from women's perceived OS without treatment. Results may have differed if we informed respondents of their chance of survival without treatment. Furthermore, the baseline levels for side-effect attributes were assumed to be the minimum possible realistic levels, however, respondents may have implicitly considered unique individual baselines based on lived experience. The interpretation of the no treatment option may have differed between respondents and may have caused some

attribute levels to appear acceptable for respondents who considered them to be the same as baseline, potentially dampening their overall effect within the sample. Results may be more precise if we estimated preferences within a more sophisticated design that adjusted for respondents' baseline levels. Finally, while the insignificance of the risk of UHA may be a genuine preference, the result may also reflect a difficulty in understanding this attribute. Despite low relative importance, similar attributes are significant in other metastatic cancer DCEs, however, the attribute levels are more severe.[37 48] Future work should explore explaining this attribute.

In conclusion, our results provide evidence that patients are willing to give up some survival benefits to avoid severe levels of side effects. Future therapeutic studies should ensure such data is collected to ensure that the patient can make an informed decision when making treatment decisions. Future research should explore using such information within a shared decision-making framework.

**Correction notice**  This article has been corrected since it was published. Licence updated to CC BY on 2nd August 2024.

**Acknowledgements**  We thank all the patients and volunteers who agreed to participate in this study and the study contributors.

**Contributors**  PH, EG, HE and MR contributed to the conceptualisation of the project and funding application. All authors contributed to the design of the study. AB developed the discrete choice experiment, with experimental design support from MR. AB and MM collected the data. AB conducted the statistical analysis with support from LEL-R. All authors contributed to the interpretation of the data as well as the drafting and revision of the manuscript. AB is the guarantor and accepts full responsibility for the finished work and/or the conduct of the study, had access to the data, and controlled the decision to publish. All authors gave final approval.

**Funding**  This research was supported by a charitable grant from the Edinburgh and Lothians Health Foundation (Scottish Registered Charity No: SC007342) and was partially supported by the Cancer Research UK (Scotland Centre CTRQQR-2021\100006). MR and LR are funded by the University of Aberdeen and the Chief Scientist Office of the Scottish Government Health and Social Care Directorates. The funders had no role in considering the study design or in the collection, analysis or interpretation of data, the writing of the paper or the decision to submit the article for publication.

**Competing interests**  None declared.

**Patient and public involvement**  Patients and/or the public were involved in the design, or conduct, or reporting, or dissemination plans of this research. Refer to the Methods section for further details.

**Patient consent for publication**  Not applicable.

**Ethics approval**  This study involves human participants and was approved by National Health Service North of Scotland Research Ethics Committee (REC ref: 19/NS/0066). Participants gave informed consent to participate in the study before taking part.

**Provenance and peer review**  Not commissioned; externally peer reviewed.

**Data availability statement**  Data are available in a public, open access repository. The data set used for this analysis is available from the University of Edinburgh Datashare, https://datashare.ed.ac.uk/handle/10283/4436.

for any error and/or omissions arising from translation and adaptation or otherwise.

**ORCID iDs**
Alistair Bullen http://orcid.org/0000-0002-1655-6404
Luis Enrique Loría-Rebolledo http://orcid.org/0000-0002-1391-6478

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
