## [Reviewer comments · BMJ Open]

ARTICLE DETAILS

TITLE (PROVISIONAL)	Trade-offs between overall survival and side effects in the treatment of metastatic breast cancer: eliciting preferences of patients with primary and metastatic breast cancer using a discrete choice experiment
AUTHORS	Bullen, Alistair; Ryan, Mandy; Ennis, Holly; Gray, Ewan; Loria-Rebolledo, Luis Enrique; McIntyre, Morag; Hall, Peter

VERSION 1 – REVIEW

REVIEWER	Gonzalez, Juan Duke University School of Medicine
REVIEW RETURNED	25-Aug-2023

GENERAL COMMENTS	Thank you for the opportunity to review the manuscript titled “Women’s preferences for overall survival versus avoiding side effects in the treatment of metastatic breast cancer: a discrete choice experiment.” The manuscript tackles an important topic that is increasingly relevant with the growing number of oral medications in this space. The risk that patients “manage” their medication intake is higher than ever. Thus, understanding patient preferences for the outcomes of these medications is key. While I commend the authors for the work done, I have some questions/concerns that should be addressed before the manuscript is considered for publication. I summarize my questions/concerns below. While I understand why the authors standardized the levels of fatigue assuming a range between grade 1 and 2. Their choice seems potentially problematic because it is not adjusted for current fatigue symptoms. It is possible that for some respondents either level of fatigue was actually an improvement over their current status, or both are actually worse than their current status, or perhaps something in between. This makes the attribute levels potentially different for every respondent. If the authors collected information about the respondent current level of fatigue, they should consider controlling for such baseline effects in the analysis. This is technically true of all other adverse effects of treatment, but I expect it to be particularly a problem with fatigue, which by the authors own admission is expected to be experienced because of the disease itself. I’m also a bit concerned about the definition of some of the side effects. Based on the definitions provided in the manuscript and the decision context, the duration of these side effects seems unclear. While the context states that they should expect to experience side effects “for weeks,” it is not clear how often they can expect to experience them. Will the side effects completely
--

disappear after those few weeks, or should they expect that the side effects would return as long as they take the treatment? Any additional information the authors could provide to clarify this, both in terms of other information provided to patients in the survey and/or evidence that whatever they meant was well understood during the pretest interviews, would be helpful.

Why both one-year and five-year survival? This seems to complicate the presentation of the attribute without any real gains in information.

Under the description of the experimental design the authors state the D-efficient design ensures minimal variation around parameter estimates. Probably better to say this minimized the variance-covariance of the average preference measures. As stated, it gives the impression that the design suppressed preference heterogeneity.

It seems OS and the risk of hospitalization were specified to be continuous AND linear. Was this tested formally? This information should be provided.

Also, I think the specification should consider a series of interaction terms:

1. Between UHA and OS as UHA carries a chance of death, so the effective expectation of survival is in fact changing with the probability of hospitalization.

2. Between the opt-in constant (Treat) and baseline expected survival. I'd expect to see more people opting in for treatment when they are more pessimistic about their prognosis without treatment. With the current model specification, this potential effect is attributed to the treatment characteristics which could dampen the negative impact of treatment side effects. This could help explain the lack of significance of nausea and diarrhea.

Also, for the evaluation of the impact of dominance, the authors should focus on the impact that these respondents have on preferences for OS—after all, these respondents are not providing information on any other attribute. This should be done by adding an interaction term in the model between a variable indicating dominance and the variable for OS. This interaction term would capture the adjustment in the value of OS for the dominant group. A simple test of significance for this parameter would suffice to determine whether this group's preferences for OS are systematically different from the rest of the sample. I do not mean to imply that the authors should eliminate these respondents if their preferences for OS are different. However, it would be important to separate the preferences of those who traded from those who didn't. The latter may be using a simplifying heuristic and so the average preferences without their influence could provide a type of sensitivity analysis.

While I understand that the authors are dealing with a small sample size, I strongly encourage them to consider (control) for preference heterogeneity to avoid any of the potential biases that can emerge from not controlling for these variations. As suggested above, variations in patients' baselines alone might affect the distribution of preferences for certain outcomes. With +100 you

	should be able to run a random-parameters logit model, even if only for the variables that are expected to be more susceptible to variations in preferences (e.g., the opt-in variable and fatigue).
--	--

REVIEWER	Goto, Yuka St Marianna University School of Medicine
REVIEW RETURNED	03-Sep-2023

GENERAL COMMENTS	This study is about using the DCE technique to identify the benefits and risks of drug treatment for patients with metastatic breast cancer, and it is clear to see that a lot of hard work has been to determine the attributes and options chosen for the DCE. However, several points as indicated below need to be addressed by authors to improve the quality of the article.  1. There appear to be several studies that have used DCE to assess preferences for drug treatment for patients with metastatic breast cancer (Page6 Line93-94). What is the novel and purpose for this study? If the novel restriction of this study is that it is the first study in the United Kingdom population, please discuss the characteristics of preferences that are unique to the United Kingdom population. 2. What was the reasons for all levels being grade 2 for the additional side effects attribute? (Page 7 Line 137-138) 3. It is known that the life risk level varies depending on the organ metastasized Lymph node and bone metastases have a low life risk and a long OS. However, metastases to the brain, lungs and liver require intensive drug therapy. The balance between OS and PFS gain and the severity of side effects in these two groups is different. Please consider adding a breakdown of the metastatic organs for 72 of the 101 patients with metastatic breast cancer included in this study to Table A2. 4. This is a study restricted to breast cancer patients. If so, compared to other DCE studies, are there any trends in preferences that are characteristic of being female, a younger group (in the 50-59 age range)? Much of the discussion section is devoted to describing the results of previous studies. Instead, how do the results of this study differ from those of previous studies? Or are they the same? 5. When conducting the survey, who was the member of staff responsible for the interview? The skill, knowledge, and relationship of mutual trust between the researcher and the patient may influence the outcome of the interview responses. Also, please describe any points that you considered during the interview, such as attention to inducements that could lead to bias.
---

REVIEWER	Wang, Aiping The First Affiliated Hospital of China Medical University
REVIEW RETURNED	04-Sep-2023

GENERAL COMMENTS	I am very honored to accept this invitation to review this manuscript, which is very well written.  1. The introduction is written logically. However, I consider that it would be better if the author emphasized the importance and necessity of this study. 2. The research design section is written in detail and the results are sufficient.
---

	3. The discussion is well written. I think it should be discussed more clearly in relation to the research results. 4. All in all, I think this article is well written and the topic is valuable.
--	--

a

VERSION 1 – AUTHOR RESPONSE

Reviewer #1	
Comment	Response
General comments	
Thank you for the opportunity to review the manuscript titled “Women’s preferences for overall survival versus avoiding side effects in the treatment of metastatic breast cancer: a discrete choice experiment.” The manuscript tackles an important topic that is increasingly relevant with the growing number of oral medications in this space. The risk that patients “manage” their medication intake is higher than ever. Thus, understanding patient preferences for the outcomes of these medications is key. While I commend the authors for the work done, I have some questions/concerns that should be addressed before the manuscript is considered for publication. I summarize my questions/concerns below.	
Specific Comments	
While I understand why the authors standardized the levels of fatigue assuming a range between grade 1 and 2. Their choice seems potentially problematic because it is not adjusted for current fatigue symptoms. It is possible that for some respondents either level of fatigue was actually an improvement over their current status, or both are actually worse than their current status, or perhaps something in between. This makes the attribute levels potentially different for every respondent. If the authors collected information about the respondent current level of fatigue, they should consider controlling for such baseline effects in the analysis. This is technically true of all other adverse effects of treatment, but I expect it to be particularly a problem with fatigue, which by the authors own admission is expected to be experienced because of the disease itself.	We accept this point; indeed, the results would be more robust if we had adjusted for the respondent baseline fatigue. Unfortunately, we did not collect data on respondents’ current symptoms. Our use of grade 1 fatigue as a base case was based on consultation with clinicians. We operate under the assumption that grade 1 fatigue represents an average baseline level for the cohort and any effect arising from differences in respondents’ baseline fatigue should be averaged out over the sample. Nonetheless, the choice framework would indeed be strengthened if it included a more clearly defined baseline fatigue for the opt-out choice or even a personalised opt-out as the reviewer has suggested. We have now elaborated further on the shortcomings of the no-treatment option in the discussion.
I’m also a bit concerned about the definition of some of the side effects. Based on the definitions provided in the manuscript and the decision context, the duration of these side effects seems unclear. While the context states that they should expect to experience side effects “for weeks,” it is not clear how often they can expect to experience them. Will	We recognise this as a potential inefficiency of the choice framework. We intended that participants would deduce from the three statements “You will still experience a side effect for weeks at a time”, “We ask you to imagine that no other treatment options will become available to you in the future”, and “Both treatments can treat you for the rest of your life” that presentation of these side effects

the side effects completely disappear after those few weeks, or should they expect that the side effects would return as long as they take the treatment? Any additional information the authors could provide to clarify this, both in terms of other information provided to patients in the survey and/or evidence that whatever they meant was well understood during the pretest interviews, would be helpful.	would be an ongoing feature of the patients' life assuming that they remained on the treatment. We were interested in the preferences for side effects which would present throughout the palliative care treatment plan. This required us to simplify the choice context so that patients would remain on the same palliative treatment indefinitely, yet we recognised during discussions with the clinicians that it would be infeasible for patients to endure side effects continuously with no respite. We therefore settled for the choice of side-effects presenting "for weeks". We wanted to represent uncertainty in how the side effects manifested but did not want to complicate the choice scenario for respondents by introducing specific risk values. We have recorded no feedback from think-aloud pilots regarding the issue of how long and when side effects presented, however, the motivation of these pilot sessions was to allow respondents to naturally provide their feedback on the survey with minimum intervention from interviewers. Therefore, pointed questions regarding participants' understanding of the choice framework were not asked. All of this being said we understand that there is some debate about how readers should interpret the results and we now acknowledge this in our discussion.
Why both one-year and five-year survival? This seems to complicate the presentation of the attribute without any real gains in information.	Our review of the qualitative literature suggested that patients' survival considerations extend to the long term. We wanted to include a constant risk variable that could be used to calculate MRS and we did not believe that the typical respondent would have an innate understanding of how a constant annual risk translates into long-term risk, this is supported by the research into the public's comprehension of risk by researchers such as David Spiegelhalter. Additional clarification has now been provided in the methods section.
Under the description of the experimental design the authors state the D-efficient design ensures minimal variation around parameter estimates. Probably better to say this minimized the variance-covariance of the average preference measures. As stated, it	Thank you for the suggestion. This has now been changed.

gives the impression that the design suppressed preference heterogeneity.	
It seems OS and the risk of hospitalization were specified to be continuous AND linear. Was this tested formally? This information should be provided.	On the reviewer's request, we have estimated an MNL specification where OS was coded as a dummy variable which has now been included in the supplementary materials. The effect of overall survival is extremely close to linear between the 60 and 75 levels used in the treatment alternatives.
Also, I think the specification should consider a series of interaction terms: 1. Between UHA and OS as UHA carries a chance of death, so the effective expectation of survival is in fact changing with the probability of hospitalization. 2. Between the opt-in constant (Treat) and baseline expected survival. I'd expect to see more people opting in for treatment when they are more pessimistic about their prognosis without treatment. With the current model specification, this potential effect is attributed to the treatment characteristics which could dampen the negative impact of treatment side effects. This could help explain the lack of significance of nausea and diarrhea.	 1. We factored this interaction into the experimental design and during analysis it had no significant effect on choice, so this was omitted from the final regression. 2. Thank you for this comment. This highlights a limitation in our dataset and sample. We agree that baseline expected survival will likely influence the preference to opt into a treatment. Likewise, it could be that respondents with low baseline survival opting into treatment would expect to suffer side effects, which could explain the lack of significance of the lesser toxicities. Unfortunately, we do not have the statistical power to test the effect of the baseline survival and these interactions. We do however now present an errors component model which, implicitly, accounts for the panel structure of the data and, crucially to this point, presents a normally distributed alternative specific constant describing treatment. Our results show significant preference heterogeneity for this variable, which is indicative of unobserved effects (e.g., the baseline survival). We have added this to the limitations section
Also, for the evaluation of the impact of dominance, the authors should focus on the impact that these respondents have on preferences for OS—after all, these respondents are not providing information on any other attribute. This should be done by adding an interaction term in the model between a variable indicating dominance and the variable for OS. This interaction term would capture the adjustment in the value of OS for the dominant group. A simple test of significance for this parameter would suffice to determine whether this group's preferences for OS are systematically different from the rest of the sample. I do not mean to imply that the authors should eliminate these respondents if their preferences for OS are different. However, it would be important to separate the	The non-traders' preferences for overall survival relative to the traders' preferences are reflected by the relative importance estimates presented in supplementary material 3. As expected the non-trader group has a stronger preference for overall survival. We believe this to be sufficient to demonstrate the relationship and did not factor the recommended interaction term into the experimental design so did not estimate a model using the term as the reviewer suggests. Thanks to the reviewer's comment we recognise that 'simplifying heuristic' is a helpful term to help readers understand the potential problems with the 'non-trader' group and now use the term.

preferences of those who traded from those who didn't. The latter may be using a simplifying heuristic and so the average preferences without their influence could provide a type of sensitivity analysis.	
While I understand that the authors are dealing with a small sample size, I strongly encourage them to consider (control) for preference heterogeneity to avoid any of the potential biases that can emerge from not controlling for these variations. As suggested above, variations in patients' baselines alone might affect the distribution of preferences for certain outcomes. With +100 you should be able to run a random-parameters logit model, even if only for the variables that are expected to be more susceptible to variations in preferences (e.g., the opt-in variable and fatigue).	As a response to the reviewer's comments, we have now replaced the main results with estimates from an errors component logit which drops the irrelevant alternatives assumption. We found this to be a suitable solution which was able to provide some indication of preference heterogeneity without compromising the power of our estimates. The BIC estimate of 836.7 indicates that the Errors-Component logit is a better fit than the MNL which has a BIC of 933.6.
Reviewer #2	
Comment	Response
General corrections	
This study is about using the DCE technique to identify the benefits and risks of drug treatment for patients with metastatic breast cancer, and it is clear to see that a lot of hard work has been to determine the attributes and options chosen for the DCE. However, several points as indicated below need to be addressed by authors to improve the quality of the article.	
Specific Comments	
There appear to be several studies that have used DCE to assess preferences for drug treatment for patients with metastatic breast cancer (Page6 Line93-94). What is the novel and purpose for this study? If the novel restriction of this study is that it is the first study in the United Kingdom population, please discuss the characteristics of preferences that are unique to the United Kingdom population.	It is difficult to characterise the general preferences of the entire population. Preferences are likely to be context-specific and this is the purpose of our study. To help position our study and highlight its importance we have now also referred to the emphasis we place on side-effect severity towards the end of the introduction.
What was the reasons for all levels being grade 2 for the additional side effects attribute? (Page 7 Line 137-138)	Thank you for this point. We have now offered additional clarifications on the characteristics of the additional side effects attribute which we believe answers this question for future readers.
It is known that the life risk level varies depending on the organ metastasized	We acknowledge the reviewer's point. We did not collect specific data relating to the metastasis location of our population. Our motivation was to minimise the disclosivity. We already briefly

Lymph node and bone metastases have a low life risk and a long OS. However, metastases to the brain, lungs and liver require intensive drug therapy. The balance between OS and PFS gain and the severity of side effects in these two groups is different. Please consider adding a breakdown of the metastatic organs for 72 of the 101 patients with metastatic breast cancer included in this study to Table A2.	addressed this within our limitations but have elaborated further to make it clearer that the location of metastases is a potential avenue for the investigation of preference heterogeneity in future research.
This is a study restricted to breast cancer patients. If so, compared to other DCE studies, are there any trends in preferences that are characteristic of being female, a younger group (in the 50-59 age range)? Much of the discussion section is devoted to describing the results of previous studies. Instead, how do the results of this study differ from those of previous studies? Or are they the same?	The preferences we estimate are highly specific to the context of the choice framework we present to the respondents. Beginning to estimate the differences between age groups for our study would require a larger sample and beyond the scope of the previously published DCEs we cite in this area. We have however identified a 2008 study which employed a questionnaire with found some general differences in preferences for quality of life and quantity of life, we now cite this in the discussion. We appreciate the reviewer's comment and recognise the previous approach to discussing the literature was incomplete. Comparisons between DCE studies can be difficult as we now discuss at the top of the paragraph. Additionally, we have included some additional text which we believe frames the discussion more clearly in relation to our work.
When conducting the survey, who was the member of staff responsible for the interview? The skill, knowledge, and relationship of mutual trust between the researcher and the patient may influence the outcome of the interview responses. Also, please describe any points that you considered during the interview, such as attention to inducements that could lead to bias.	Thank you for identifying this consideration. The reviewer is correct that the interviewer is likely to have a significant impact on the results of the qualitative work. The qualitative work was completed between a research nurse and a research assistant trained in qualitative methods. This information is highlighted in the supplementary materials, but we now include additional detail in the manuscript. We also provide additional detail regarding the approach to the interviews which we believe addresses the reviewer's questions regarding our methods.

Reviewer #3	
Comment	Response
General comments	
I am very honored to accept this invitation to review this manuscript, which is very well written.	
Specific Comments	
The introduction is written logically. However, I consider that it would be better if the author emphasized the importance and necessity of this study.	We have now provided further detail about our study's positioning in the wider literature as well as its contribution.
The research design section is written in detail and the results are sufficient.	Thank you. No action is required.
The discussion is well written. I think it should be discussed more clearly in relation to the research results.	We now discuss the results of our paper more clearly in relation to the results of existing published studies in the area of metastatic breast cancer treatment preferences.
All in all, I think this article is well written and the topic is valuable.	Thank you. No action is required.